# Immune Checkpoint Inhibitors in Cancer Therapy—How to Overcome Drug Resistance?

**DOI:** 10.3390/cancers14153575

**Published:** 2022-07-22

**Authors:** Yefang Lao, Daoming Shen, Weili Zhang, Rui He, Min Jiang

**Affiliations:** 1Department of Oncology, The First Affiliated Hospital of Soochow University, Suzhou 215006, China; yflao123@stu.suda.edu.cn; 2Department of Internal Medicine, Xiangcheng People’s Hospital, Suzhou 215131, China; shendaomingxc@163.com; 3Department of Gastroenterology, Xiangcheng People’s Hospital, Suzhou 215131, China; zhangweilizhang@sina.com; 4Department of Pneumoconiosis, Shanghai Pulmonary Hospital, Shanghai 200433, China

**Keywords:** immune checkpoint inhibitors, resistance, immunotherapy

## Abstract

**Simple Summary:**

Immune checkpoint inhibitors (ICIs) are an important strategy in cancer therapy. However, with the widespread clinical use of ICIs, people gradually found that ICIs may not be effective enough to eliminate tumor tissue for certain patients. The resistance to ICI treatment makes some patients unable to benefit from their antitumor effects. Therefore, it is vital to understand their antitumor and drug resistance mechanisms to better narrow the ICI-resistant patient population. This review outlines the antitumor action sites and mechanisms of different types of ICIs and lists the main reason of ICI resistance based on recent studies. Finally, we propose current and future solutions for resistance to ICIs.

**Abstract:**

Immune checkpoint inhibitors (ICIs), antagonists used to remove tumor suppression of immune cells, have been widely used in clinical settings. Their high antitumor effect makes them crucial for treating cancer after surgery, radiotherapy, chemotherapy, and targeted therapy. However, with the advent of ICIs and their use by a large number of patients, more clinical data have gradually shown that some cancer patients still have resistance to ICI treatment, which makes some patients unable to benefit from their antitumor effect. Therefore, it is vital to understand their antitumor and drug resistance mechanisms. In this review, we focused on the antitumor action sites and mechanisms of different types of ICIs. We then listed the main possible mechanisms of ICI resistance based on recent studies. Finally, we proposed current and future solutions for the resistance of ICIs, providing theoretical support for improving their clinical antitumor effect.

## 1. Introduction

Cancer is a major cause of death and a considerable obstacle to improving life expectancy in countries around the world [1]. To date, the world has approximately 20 million new cancer cases and nearly 10 million cancer deaths [2]. Cancer treatment has long been a central challenge in the fight against human disease. With the deepening of cancer research, people have gradually realized that the immune system plays a critical role in the development and metastasis of tumors [3,4]. Tumor cells can produce an immunosuppressive tumor microenvironment by releasing a large number of immunosuppressive signaling factors. This causes tumors to escape the monitoring of the body’s immune system, and even to use immune cells to promote their own growth. Immunotherapy harnesses the patient’s own immune response to trigger an antitumor effect; this has become another crucial revolutionary treatment means after surgery, radiotherapy, chemotherapy, and targeted therapy [5,6]. A large amount of data indicate that progression-free survival (PFS) and overall survival (OS) are significantly enhanced in patients receiving immunotherapy [7]. Compared with traditional cancer treatment, immunotherapy has better development prospects.

Immune checkpoints are molecules expressed in immune cells that can regulate the degree of immune activation and play an important role in preventing autoimmunity [8]. However, tumor cells “skillfully” make use of the negative regulatory effect of immune checkpoint molecules in immune cells such as T cells, macrophages, natural killer (NK) cells, and dendritic cells (DCs). The tumor tissue could avoid T-cell-mediated cytotoxic damage through high expression of checkpoint molecule ligands, thus causing immune suppression and ultimately leading to the occurrence, invasion, and even metastasis of cancer [9]. Immune checkpoint inhibitors (ICIs), antagonists that can competitively bind checkpoint molecules, such as PD-1/PD-L1 and CTLA4/B7, comprise the main antitumor strategy in immunotherapy [10]. As the most common active immunotherapy, ICIs have shown great potential in the treatment of advanced metastatic and highly immunogenic tumors such as melanoma or Merkel cell carcinoma [10,11,12]. Furthermore, several cancer treatment strategies based on immune checkpoint blocking are already in clinical trials, including blocking antibodies to CTLA-4 (Yervoy^®^; NCT02017717) and an anti-PD-1 antibody (MED14736; NCT02336165); all these trials have better antitumor effects. With the discovery of immune checkpoints and the development of an increasing number of antagonists, ICI-based immunotherapy for tumor therapy strategies would become another vital choice for tumor treatment.

However, with the extensive use of ICIs in clinical practice, the antitumor efficacy of ICIs is still inadequate. First, although the clinical antitumor effects of immunosuppressants are encouraging in some clinical trials, their efficacy against solid tumors is not as expected due to the heterogeneity of solid tumors and the external microenvironment [13]. Some clinical results showed the remission rate of patients treated with CTLA-4 inhibitor is about 15%, and the remission rate of PD-1/PD-L1 inhibitor rarely exceeds 40%. In addition, low clinical target response rates and the risk of causing autoimmune diseases are still the chief limitations of ICIs [14]. Second, according to the data, there are still some patients in the ICI tolerance state, even though ICIs have a good tumor inhibition effect, which would lead to some patients not benefitting from ICI treatment [15]. Therefore, in order to develop ICIs, it is important to determine their underlying causes of partial resistance and partially low reactivity to effectively improve the clinical returns of cancer patients. For this paper, we summarized the antitumor sites and mechanisms of ICIs. More significantly, we focused on the causes and possible mechanisms of ICI resistance in some patients. Finally, we proposed current and future solutions for the resistance of ICIs, providing theoretical support for enhancing their clinical antitumor effect.

## 2. Classification and Mechanisms of the Antitumor Action of ICIs

As checkpoint antagonists, ICIs play a primary direct role in boosting immunosuppression. With their excellent antitumor effect, they have been widely applied in clinical settings and have achieved therapeutic outcomes in a variety of cancer treatments [16]. The immune checkpoint is an important target to prevent immune cells from attacking normal tissue and plays an essential role in maintaining normal tissue homeostasis. Immune checkpoint receptor/ligand proteins are expressed in a variety of immune and normal tissue cells. In tumor tissue, to escape the monitoring and killing of the immune system, tumor cells can produce a large number of immune checkpoint proteins to make the immune system unable to recognize tumor tissues [13]. ICIs have been widely developed as a central drug to block tumor inhibition of immune cells (Table 1) [17]. ICI drugs can be transformed into different drugs targeting receptors and ligands of the same inhibitory pathway, such as PD-1 and PD-L1 inhibitors in the PD-1/PD-L1 signaling pathway. Most researchers believe there is little difference between the two inhibitors; however, they target different cell types and have produced different therapeutic outcomes and cytotoxicity in recent studies [18,19]. In this regard, ICI drugs are divided into different targeted directions for different cell types, which can be used to target tumor cells or some immune cells, thus promoting the antitumor effect (Figure 1) [20]. In this section, we summarize and classify the major immune checkpoints on different cell surfaces and clarify the key mechanisms of ICIs.

### 2.1. ICIs for Immune Cells

#### 2.1.1. ICIs for CD8^+^ T Cells

Programmed death 1 (PD-1) is an inhibitory transmembrane protein mainly expressed on the surface of the T-cell membrane and is the most classical T-cell-associated immune checkpoint [21,22]. Its chief physiological function is to bind with the ligand PD-L1 or PD-L2 to inhibit the immune toxic function of T cells, and then to recognize the body’s normal tissue cells to prevent them from experiencing toxic damage [23]. Notwithstanding, under the influence of numerous abnormal inflammatory environments in the tumor microenvironment and the abnormal overexpression of PD-1 ligand in tumor cells, the expression of the PD-1 receptor in effector T cells was significantly increased in one study, resulting in T-cell failure. In addition, activation of the PD-1 signaling pathway may transform effector T cells into regulatory T cells (Tregs), resulting in impaired antitumor immune responses [24,25]. Yokosuka et al. (2012) found that the binding of PD-1 and PD-L1 can mobilize the inhibitory phosphatase SHP2 in T cells to dephosphorylate the proximal TCR signaling molecule, thus reducing the antigen recognition of TCR proteins and causing the immune escape of tumor cells [26]. More importantly, PD-1 can indirectly inhibit the function of effector factors produced by TCR transduction [27]. PD-1 inhibitors, designed to reactivate the immunotoxic function of T cells, produce powerful antitumor effects. There have been a number of clinical trials of antitumor PD-1 drugs, and excellent cancer treatment effects have been reported (clinical trials). PD-1 inhibitors can effectively block the binding of T cells to the tumor cell immune checkpoint PD-1/PD-L1 by competing for newly binding PD-1 receptors located on the surface of T cells to reactivate the immunotoxic function of T cells and generate positive antitumor effects. Ansell et al. and Ramchandren et al. concluded, respectively, in two independent phase I/II and phase II clinical trials in 2015 and 2019, that the PD-1 antibody nivolumab showed significant therapeutic activity and acceptable safety in the treatment of relapsed and refractory classical Hodgkin’s lymphoma [28,29].

In addition, CTLA-4 (cytotoxic T lymphocyte antigen-4, CD152), as a coinhibitory receptor on the surface of T cells, is another typical T-cell-associated immune checkpoint inhibitor signaling pathway that mostly mediates tumor immunosuppression through antigen-specific signaling pathways. CTLA-4 belongs to the immunoglobulin superfamily, which induces T cells to enter a quiescent state by binding to B7 ligand located on antigen presenting cells (APCs) [30]. CTLA-4 also mediates the transendocytosis of APCs on B7-1 and B7-2, preventing their binding to CD28 and reducing the availability of CD28 [30,31]. In addition, CTLA-4 limits the residence time of the T-cell receptor (TCR) and antigen-presenting cells [32]. Moreover, CTLA-4 can lead to dephosphorylation and the inactivation of positive immune stimulation signals [33]. It can also transmit signals that directly inhibit T-cell activation, resulting in the inhibition of T-cell proliferation and activation, and the reduced release of pro-effector cytokines such as IL-12, and cytotoxic enzymes such as perforin [34].

Similarly, TIM-3, another newly discovered T-cell immune checkpoint, has been proven to be a key signaling pathway molecule in the immunosuppressive response. By binding with its ligand galectin-9, TIM-3 can further induce apoptosis or inhibit the differentiation of effector T cells, downregulate the immune response, and thus induce immune tolerance [35]. Moreover, TIM-3 has a regulatory effect on APC cells such as DC cells, which blocks the potential of promoting antitumor immunity by modulating inflammasome activation [36]. The development of a TIM-3 antibody provides another option for immune checkpoint targets targeting T cells, and even APC cells, and can be combined with PD-1 inhibitor therapy to synergistically boost the immune activation of T cells. To date, there have been a large number of clinical trials using TIM-3 for the treatment of various solid tumor patients, and some progress has been made. In addition, Curigliano et al. used sabatolimab (TIM-3 inhibitor) and spartalizumab (PD-1 inhibitor) in combination to treat a variety of solid tumors, including ovarian and rectal cancer, compared with spartalizumab alone. A better antitumor effect was observed through the combined action [37]. As T cells have the most direct killing effect on tumor cells, this has become a key research direction to explore more effective related signaling pathways inhibited by tumor tissues. Such research will become the basis for immunotherapy with ICIs.

#### 2.1.2. ICIs for Macrophages

Macrophages are central effector molecules of the innate immune system, detecting activation and inhibition signals at any time to initiate phagocytosis and the secretion of cytokines [38]. As the immune cells with the highest content of the tumor immune microenvironment, macrophages play a primary role in the occurrence and development of cancer, and also in invasion and metastasis [39]. Under the influence of tumor cells, tumor-associated macrophages not only fail to inhibit tumor growth but also promote the progression of cancer due to their inflammatory phenotype and release of related factors [40]. In recent years, immune checkpoints targeting macrophages have been developed.

SIRPα is an inhibitory receptor containing multiple intracellular tyrosine-based immunosuppressive motifs (ITIMs), which are mostly expressed on the surface of APCs, such as macrophages [41]. The ligand of TIM-3, CD47, acts as a “do not eat me” signal that protects healthy cells from becoming engulfed by macrophages; hematopoietic cells lacking CD47 are quickly engulfed by macrophages and trigger dendritic cell activation [42]. However, CD47 is highly expressed on the surface of tumor cells, which serves as a mechanism for evading immunodetection. When SIRPα binds to the ligand CD47, SIRPα is phosphorylated and then recruits phosphatases SHP-1 and SHP-2, which inhibit phagocytosis and the functional activity of macrophages [43,44]. SIRPα inhibitors targeting macrophages have emerged as a strategy to break down tumor inhibition of macrophages. In 2020, Kuo et al. used the SIRPα blocker hAB21 to foster macrophage-mediated antibody-dependent phagocytosis of tumor cells in vitro. They found that a SIRPα antibody enhanced the response of CD20-positive lymphomas to rituximab, and significantly promoted the antitumor activity of PD-1/PD-L1 inhibitors [45]. In addition, Wu et al. confirmed the effective impact of the SIRPα antibody BR105 in 2022 [46]. BR105, combined with the CD20 antibody or the HER-2 antibody in vitro, could promote macrophage phagocytosis of CD20-positive Burkitt lymphoma cells or SK-BR-3 breast cancer cells, overexpressing HER-2. In a mouse tumor model, BR105 combined with the CD20 antibody significantly inhibited lymphoma growth. The CD47-SIRPα axis plays a key role in antitumor immunity. However, SIRPα inhibitors alone were insufficient to induce strong antitumor immunity [47,48,49]. The use of a SIRP antibody as an inhibitor has positive significance for improving the antitumor effect, and SIRP inhibitors have become an important choice for collaborative therapy.

In recent years, with in-depth research on tumor-associated macrophages, several other macrophage-related inhibitory signaling pathways have been discovered. Leukocyte immunoglobulin-like receptor B (LILRB) is expressed in most immune cells, and in binding to the major ligand major histocompatibility complex I (MHCI), mediates the negative regulation of immune cell activation [50,51]. MHCI is a complex formed by the HLAα chain and β 2-microglobulin (β2 M) [52]. Anchored by glycosyl phosphatidylinositol, CD24 interacts with sialic acid, binding immunoglobulin-like agglutinin-10 (SIGlec-10) to β2 M, which is overexpressed in some tumor tissues and binds to LILRB1 on macrophages to inhibit phagocytosis, resulting in a loss of immune surveillance [53,54]. Thus, in patients with normal or high MHCI expression in tumor cells, drugs targeting the MHCI/LILRB1 axis may facilitate antitumor immune responses and act synergistically with drugs targeting the CD47/SIRPα axis. In addition, CD24, known as a heat stable antigen, is a highly glycosylated surface protein [55,56] that reduces harmful innate immune-mediated inflammation caused by infection or liver injury. In binding to CD24, SIGLEC-10 recruits and activates sh2-domain tyrosine phosphatase SHP-1 or SHP-2, thereby blocking the cytoskeletal rearrangement required for macrophage phagocytosis and triggering an inhibitory signaling cascade that ultimately leads to the inhibition of macrophage phagocytosis [57].

The presence of different immune checkpoint inhibition signals in macrophages leads to immune selection of drug-resistant cancer cell subsets of macrophages, allowing tumor cells to evade macrophage surveillance and clearance. Therefore, understanding and targeting the mechanisms of antiphagocytic signals on the surface of macrophages can better predict therapeutic effects and targeted therapies. Macrophage-related ICI drugs have also been developed in large numbers and have shown excellent clinical antitumor efficacy, providing more options for macrophage-related ICI antitumor activity.

#### 2.1.3. ICIs for NK Cells

The NK cell is a kind of cytotoxic lymphocyte, which also plays an important role in the antitumor innate immune system. Recent studies have found that there are also immune checkpoints between NK cells and tumor cells, which could also be immuno-suppressed by tumor cells. The antitumor function of NK cells depends on the integration of activation and inhibition signals. The specific inhibitory receptors in NK cells mainly include killer cell immunoglobin-like receptor (KIR), natural killer Group 2A (NKG2A), and leukocyte immunoglobin-like receptor (LIR). All of these three molecules are regulated by MHC-Ι on the surface of tumor cells. KIR is a member of the immunoglobulin superfamily, which could recognize classical human leukocyte antigens A, B, and C (HLA class Ia) molecules. People have successively demonstrated that KIR inhibitor can enhance the antitumor effect of NK cells in vitro and in vivo [58,59]. Meanwhile, many clinical trials of KIR inhibitor are also ongoing (NCT01256073) with excellent inhibition of tumor growth without significant adverse reactions in patients with first complete remission [60]. In addition, NKG2A is a member of the C-type lectin family of receptors, which can dimerize with CD94 to form NKG2A/CD94 receptor and recognize non-classical HLA-E class I molecule. NKG2A is also one of the important immune checkpoints. In 2018, Pascale Andre´ et al. found that blocking NKG2A in vitro could promote the immune response of NK cells. More importantly, the combination of anti-NKG2A antibody and anti-PD-L1 antibody can further improve the control of tumor growth. Meanwhile, they also confirmed the safety and efficacy of Monalizumab (anti-NKG2A antibody) in combination with cetuximab in a Phase II trial (NCT02643550) for re-lapsed or metastatic squamous cell carcinoma of the head and neck therapy [61]. LIR is also known as ILT (immunoglobin-like transcript), which can mainly recognize non-classical HLA-G (class Ib) molecules. Previous studies also proved that LIR-1 blocking combined with lenalidomide could restore NK cell function in chronic lymphocytic leukemia [62]. 

Interestingly, all these three inhibitors recruited phosphatase SHP-1/SHP-2 through ITIMs to exert immunosuppressive effect [63]. Therefore, when tumors downregulated MHC-I molecules to avoid CD8+T cells, NK cells could then recognize and kill them due to the weakening of inhibition signal by all the KIR, NKG2A, and LIR molecules and play an antitumor effect [64]. From this perspective, activation of NK cells is another important antitumor compensation mechanism for T-cell immunosuppression. Immuno-antitumor strategies on NK cells would have stronger therapeutic potential.

#### 2.1.4. ICIs for DCs

DCs, as the main kinds of APCs, play an active role in the uptake, processing, presentation, and activation of T-cell function by tumor-associated antigens. Immunotherapy research aimed at activating DCs has made significant progress in clinical studies. Although DCs have no ability to kill tumor cells directly, they play a crucial role in tumor recognition. With the development of immunotherapy, it has been found that DCs are also immunosuppressed by tumor cells in the TME. In 2016, Carmi et al. found through a series of experiments that the Src homology region 2 domain-containing phosphatase-1 (SHP-1) and phosphatase-regulating regions of Akt activation are closely related to the antitumor effect of DCs. The results showed that in the TME with a weak DC response, the levels of SHP-1 and Akt phosphorylase were significantly increased. Concurrent application of the SHP-1 inhibitor phosphatase and tensin homolog (PTEN) can effectively activate DCs and the immune response. Hence, SHP-1 and Akt phosphorylation may serve as important immune checkpoints for DCs to exert antitumor effects [65]. Yuan et al. (2022) revealed that vitamin E could restore DC function in the tumor microenvironment and enhance antigen presentation and antitumor immune effects by binding and inhibiting SHP-1 [66]. In addition, the JAK/STAT3 signaling pathway and BTK-IDO signaling pathway have a significant inhibitory effect on DCs. Inhibition of these two signaling pathways in the TME by STAT3 inhibitors or indoleamine 2,3-dioxygenase (IDO) inhibitors can also improve the antitumor activity of DCs [67,68,69]. In sum, DCs, as an important component of the tumor immune microenvironment, play an indelible role in tumor recognition and T-cell activation.

### 2.2. ICIs for Tumor Cells

In the process of tumor tissue development, to imitate normal tissue cells from immune system monitoring and clearance, the tumor cell membrane surface often exhibits a variety of abnormally high expressions of immune checkpoint ligand proteins [70]. These immune checkpoints target T cells, APCs, and other immune cell types; release incorrect signals; and inhibit immune activity, leading to immune escape of tumor cells [8]. Different ICIs have been developed for immune checkpoint signaling pathway ligand receptors. These ICIs act on the ligand or receptor end of these inhibitory signaling pathways and have a positive effect on the blocking effect of the pathway [10]. However, with the development of research, it has been found that different antibodies targeting immune cells and tumor cell membrane surfaces have certain differences in efficacy and toxicity. For the same checkpoint pathway, it is more effective to directly activate immune cells by blocking T cells and other immune cell membrane surface receptors. Likewise, due to the specificity of generalized immune cells, ICIs targeting immune cells may induce systemic immune activation, which may cause unique toxic effects of the immune system, such as cytokine release syndrome (CRS) [71]. Blocking the surface ligand of the tumor cell membrane can cut off the inhibitory effect of tumor tissue on the immune system and then promote the activation of immune cells. Although this approach does not induce stronger T-cell activation, ICIs only target a subset of tumor cell subsets with high expression of abnormal checkpoint ligands, which may lead to ICI resistance. However, this milder blockade has better biosafety [72,73]. Therefore, for different cancer patients, the personalized selection of an ICI treatment strategy is an important means of decision making.

### 2.3. Other Emerging Immune Checkpoints

As the research on immune checkpoints deepens, an increasing number of new immune checkpoints are being discovered, which undoubtedly generates more options. Lymphocyte activation gene 3 (CD223) is an immune molecule widely expressed in activated T cells, NK cells, B cells, and plasmacytoid dendritic cells [74]. Its extracellular domain has 20% amino acid homology with CD4. As such, LAG-3 is closely tied to CD4 but has different functions [75]. In 2018, Maruhashi et al. found that LAG-3 did not generally recognize MHCII, but selectively recognized stable peptide-MHCII binding (pMHCII) [76]. In addition, LAG-3 and PD-1 may play a strong synergistic role in tumor immunosuppression. In 2012, Woo et al. found that PD-1 and LAG-3 were widely expressed in tumor-infiltrated CD4^+^ and CD8^+^ T cells. In transplanted tumor mice, the anti-LAG-3 antibody combined with the anti-PD-1 antibody could treat the vast majority of mice, prolonging their survival period and improving tumor removal ability [77]. Subsequently, the combined antitumor effect of a LAG-3 inhibitor and PD-1 inhibitor was found in mouse tumor models of ovarian cancer and chronic lymphocytic leukemia [78,79]. These results suggest that LAG-3 and PD-1 have a strong synergistic effect on the regulation of T-cell function and immunosuppression, which may play a powerful role in tumor therapy, and the double blocking of the two has considerable prospects. Several immunotherapy options for LAG-3 are already in clinical trials (NCT01968109 or NCT03489369) and have demonstrated excellent clinical efficacy.

TIGIT is mainly expressed on the surface of activated T cells and NK cells, and its receptors are primarily poliovirus receptors (PVR, CD155) and VR-like protein 2 (PVRL2, CD122), which are highly expressed in tumor cells and APCs, and are newly discovered immune checkpoint ligands. TIGIT induces massive IL-10 production by binding to CD155 on APCs, which reduces il-12 production and indirectly weakens the T-cell response [80,81]. In 2014, Johnston et al. found that TIGIT could directly interact with the costimulator CD226 on the cell’s surface, which destroyed the homologous dimerization of CD226 and weakened the activation and stimulation of T cells, indicating that this checkpoint plays a vital role in inhibiting T-cell function. In addition, the study further revealed that in transplanted tumor mice, the simultaneous blockade of TIGIT and PD-1 significantly inhibited tumor growth, and only the dual blocking mice produced significantly increased IFN-γ in tumor-infiltrating CD8^+^ T cells, implying a synergistic role of TIGHT and PD-1 in T-cell suppression [82]. In addition, Zhang et al. (2018) found that TIGIT deficiency in NK cells alone can cause significant tumor growth delay in mice, and anti-TIGIT antibodies can reverse NK-cell function depletion in various tumor models, indicating that NK cells in TIGIT are closely linked to tumor immunity [83]. As they are potential immune checkpoint targets, many clinical trials on TIGIT inhibitors as single agents and in combination with PD-1 are underway.

VISTA, a member of the B7 family, is one of the key immune checkpoints discovered in recent years. Unlike most immune checkpoints, VISTA is not expressed in activated T cells but rather in naive T cells. When VISTA is absent or a specific antibody is blocked, naive T cells are more likely to be activated by TCR and cytokines, resulting in a stronger response and a significant expansion of antigen-specific T cells and reduced tolerance [84]. VISTA may function both as a receptor and ligand in the immune environment [85,86]. PSGL-1 and VSIG3 have been established to bind VISTA to exert immunosuppressive effects, and their binding is regulated by pH. VISTA binds to VSIG3 at physiological pH, and PSGL-1 binds to VSIG3 in the vast majority of the tumor microenvironment (TME) at pH < 6. Therefore, to target the TME, pH may become a hotspot of VISTA research in the future [86,87,88]. For example, Wang et al. (2011) found that VISTA was highly expressed in APCs and CD4^+^ T cells in a mouse model, and could inhibit T-cell effects for a long time. Inhibition of the VISTA antibody exacerbates T-cell-mediated autoimmune disease in mice [89]. Lines et al. (2014) described the structure, expression, and function of human VISTA, which has a negative immune regulation similar to that in mice [90]. Studies on topics such as the phase 1 trial of VISTA inhibitor (CI-8993) in advanced solid tumors (NCT04475523), VISTA inhibitor alone (HMBD-002), or pembrolizumab (PD-1 inhibitor) in combination in advanced solid tumors (NCT05082610) are in progress.

## 3. The Mechanisms of Resistance to ICIs

Immunotherapy using ICIs is a mature method and has been widely used in clinical settings. However, according to clinical data, drug resistance still exists in some patients treated with ICIs [91]. Some patients who undergo ICI treatment fail to respond to antitumor therapy [92,93]. T cells are the most important terminal killer cells targeted at cancer cells in immunotherapy. Their killing effect on tumors requires three essential steps: high infiltration, the activation of T cells, and correct recognition of tumor cells. That said, in the complex tumor microenvironment, it is difficult to implement all three requirements by using ICIs. Resistance to ICIs is associated with multiple factors in the complex tumor microenvironment [13,94,95]. It is only by removing obstacles to the role of ICIs through individualized combination therapy that ICIs can produce better clinical efficacy. This section will summarize the reasons for resistance to ICIs in certain clinical cases.

### 3.1. The Complexity of the Immune Microenvironment

The tumor microenvironment is composed of immune cells, fibroblasts, inflammatory cells, surrounding vascular tissue, signaling molecules, and an extracellular matrix [96,97]. Among them, the suppressed immune system plays a crucial role in promoting the occurrence and development of tumors. However, the heterogeneity of tumors is complex and variable in different patients, different sections of tumors in the same patient, and different parts of tumors in the same patient [98]. For the immune microenvironment, there are great differences in immune cell types, sensitivity, and degree of invasion, resulting in different efficacies of ICIs. The status of the immune microenvironment may directly determine resistance to ICIs.

#### 3.1.1. Low Immune Infiltration in Tumor Tissue

ICIs could block the function of inhibition between tumor and immune cells, thereby increasing the destruction of T cells to kill tumor cells. Therefore, the premise of immune-antitumor therapy is the presence of a large number of suppressed immune cells, such as T cells. T cells in the TME play the most important role in immune surveillance and tumor killing, and the therapeutic effectiveness of ICIs is mainly driven by activated T cells. However, T-cell-dominated immune cells in some tumor microenvironments are very poorly infiltrated, which often results in the inability to re-establish a strong antitumor immune response after ICI treatment. As early as 2003, Hodi et al. explored the biological activity of the anti-CTLA-4 antibody in patients with metastatic melanoma or ovarian cancer. A large number of T cells were infiltrated in the tumor biopsies of some patients, including CD8^+^ T cells and CD4^+^ T cells [99]. Ribas et al. further analyzed tumor biopsies of melanoma patients who received anti-CTLA4 antibodies in 2009. They found that clinically responsive tumors had a significant increase in the intracellular diffuse infiltration of CD8^+^ T cells compared to baseline, with or without an increase in the CD4^+^ T cells. However, there was only sparse patchy CD8^+^ T cells infiltration in the unsubsided lesions [100], indicating that the CTLA-4 inhibitor treatment response was related to T-cell infiltration in the TME. In addition, Tumeh et al. (2014) demonstrated that PD-1/PD-L1 immunosuppressants work only after the presence of CD8^+^ T cells in the tumor. By scrutinizing tumor samples from patients with metastatic melanoma before and during treatment with pembrolizumab, they found that patients who responded to PD-1 inhibitor treatment had a large extent of CD8^+^ T cells infiltration at stroma-tumor margins, while drug-resistant patients displayed significantly low CD8^+^ T cells infiltration [101]. Hence, low immune infiltration in the TME may lead to a non-response to ICI treatment.

#### 3.1.2. Low Immunogenicity in Tumor Tissue

When ICIs enter the TME, the T-cell inhibitory pathway by tumor cells is cut off, and T cells are activated. Notwithstanding, T-cell destruction of tumor cells requires correct recognition of target cells. The lack of immunogenicity directly diminishes T-cell recognition of tumors, which would render activated T cells unable to produce a precise tumor killing effect. Immunogenicity depends on two key factors: antigenicity and adjuvanticity. Antigen polypeptides on the surface of tumor cells bind to MHC molecules on T cells, which causes tumor cell stress or death and then releases adjuvant signals such as damage-associated molecular patterns (DAMPs) to produce an effective immune response [102,103].

Tumors with a high response to ICIs tend to have a high mutation load, which also means that the tumor has strong immunogenicity. This would make it easier for T cells to target tumor cells to generate effective antitumor immunity. Le et al. (2015) found that the objective response rate of PD-1 inhibitor treatment could reach 40% for colorectal cancer patients with a high mutation load of mismatch repair deficiency (dMMR) through clinical trials, while for mismatch repair proficient (pMMR) patients, the objective response rate was 0 [104]. Additionally, Anagnostou et al. (2017) found that in patients who developed acquired resistance during ICI therapy for NSCLC, the loss of tumor antigens was associated with the loss of T-cell recognition in peripheral blood. Coculture of the missing antigens with autologous T cells would lead to the expansion of T cells, which can induce a functional immune response by tumor antigens [105]. The initial low immunogenicity of tumor cells could result in insufficient recognition of T cells, leading to primary drug resistance. The loss of key tumor antigens during ICI therapy, such as the weakening of antigens related to TCR or MHC recognition, may lead to secondary drug resistance. As such, the existence of tumor antigens and the continued recognition of T cells are important factors for the curative effect of ICIs. Ensuring a continuously high concentration of antigens is one of the chief strategies to overcome ICI resistance in some cancer patients.

### 3.2. Intersecting Immunosuppressive Pathways

As tumors develop, genetic mutations accumulate in cancer cells, leading to the overexpression of more than one immune checkpoint ligand on the surface of the tumor cell membrane. Hence, there is often more than one immune checkpoint signaling pathway that prevents immune cells from functioning. This cross-existence of immunosuppressive pathways makes tumor immune escape more difficult to reverse and thus becomes a central mechanism of ICI resistance. Multiple immune checkpoints in the tumor immune evasion system can synergistically inhibit multiple immune cells; that is, the mechanism of ICI resistance is linked to the clustering effect of the checkpoint network. For example, as a classical immune checkpoint of T-cell dysfunction, PD-1 inhibitors have positive therapeutic significance for cancer patients. However, significant resistance to PD-1 inhibitors still exists in some cancer patients. Koyama et al. (2016) found that PD-1 inhibitor resistance may be associated with the upregulation of other immune checkpoints. In particular, TIM-3 expression was increased in EGFR- and KRAS-driven lung adenocarcinoma mouse models, which progressed after PD-1 inhibitor treatment. Patients exhibited a better survival advantage when TIM-3 inhibitors were added, and similar results were observed in two patients who developed adaptive resistance to PD-1 inhibitors [106]. The combination of TIM-3 and PD-1 inhibitors may generate better tumor suppression and reduce the number of resistant populations.

In addition, LAG-3, TIGIT, and PD-1 are co-expressed on the surface of T cells and are all immune checkpoint ligands subject to tumor immunosuppression. Studies of LAG-3 and TIGIT checkpoints revealed that the combination of LAG-3 or TIGIT inhibitors and PD-1 inhibitors often had a synergistic effect of “1 + 1 > 2.” This indicates that the three signaling pathways have some cross-drug resistance to the inhibition of T-cell function [77,78,79,82,107]. As such, the complex checkpoint compensatory upregulation or dynamic change in resistance mechanisms in the immune microenvironment is an important research direction to overcome the progression of ICI resistance. The blocking effect of dual (or even multiple) inhibitors may produce better synergistic immune activation.

### 3.3. Other Mechanisms of Tumor Resistance

#### 3.3.1. Route and Dose of ICI Administration

In the tumor microenvironment, the concentration of ICIs often directly determines the degree of activation of T cells and affects antitumor efficacy. However, for solid tumors, physical, biological, and chemical barriers prevent ICIs from accumulating around the tumor. In clinical settings, intravenous injection is often the primary method of drug administration. ICIs are initially diluted with the continuous flow and dispersion of blood, and only a few inhibitors can actually enter the tumor microenvironment and play a positive role [108]. Not only do these inhibitors fail to produce effective and sustained functional activation of immune cells, but they may also induce tumor cells to “learn and renew” themselves, producing more forms of immune escape and developing drug resistance. With the development of research, how to break through the three tumor barriers to improve the targeting of ICIs has become a new antitumor immunosuppressive method. The concept of drug delivery has been proposed to increase the concentration of ICI drugs in the tumor microenvironment. For example, membrane-wrapped drugs or cell types with homing functions were first used to deliver drugs more accurately to tumor sites and improve local concentration [109].

Further, with the breakthrough of nanotechnology, good biocompatibility and excellent targeting of nanomaterials have been discovered. Using organic or inorganic nanomaterials to deliver ICI drugs has become an important form of drug delivery. These nanodrugs first accumulate in tumors via passive/active targeting mechanisms and then further develop their therapeutic efficacy [110,111]. For example, Cheng et al. (2018) synthesized therapeutic peptide assembly nanoparticles coated with PD-L1 inhibitors and IDO inhibitors for melanoma therapy. Among them, functional 3-diethylaminopropyl isothiocyanate (DEAP), a peptide substrate of matrix metalloproteinase-2 (MMP-2), is the main component of nanomaterials. DEAP is protonated in a weakly acidic tumor environment, and the nanostructure expands, releasing the correlation and achieving a therapeutic effect of controlling tumor progression [112]. Chen et al. (2019) incorporated calcium carbonate nanoparticles loaded with anti-CD47 antibodies (aCD47@CaCO3) into fibrin gels (aCD47@CaCO3@fibrin) and sprayed them onto post-tumorectomy mice. It was found that the material could promote the polarization of TAMs to the M1 phenotype, and immunosuppressive cells such as MDSCs and Tregs were reduced. Meanwhile, the release of the anti-CD47 antibody enhanced the phagocytosis of macrophages to tumor cells and inhibited local tumor recurrence and potential metastasis and diffusion after surgery [113]. Therefore, nanotechnology has many advantages of achieving the precise controlled release of drugs, improving local drug concentration and drug bioavailability, and reducing adverse drug reactions.

#### 3.3.2. Other Factors of ICI Administration

The presence of highly expressed immune checkpoint signaling pathway proteins on the surface of tumor cells and T cells is responsible for the strong inhibitory effect of tumors on the immune system. In addition, long-term stimulation of tumor-associated antigens results in T-cell depletion and the secondary overexpression of immune checkpoint signaling proteins, resulting in further suppression of immune function [114]. Pauken et al. (2016) found that the failure of PD-1 inhibitors to maintain long-term antitumor effects and T-cell activation may be related to the stability of T-cell epigenetic inheritance [115]. Meanwhile, Ahn et al. found that in the PD-1/PD-L1 checkpoint pathway, the expression level of PD-1 in T cells is mostly driven by demethylation of the pD-1 promoter [116]. T cells with a stable demethylation drive appear to have ICI resistance. This epigenetic stability leads to another possible explanation for tumor resistance, which is the inability to maintain long-term antitumor effects and the inability to activate T cells after the use of PD-1 inhibitors [115]. Hence, the combined use of epigenetic modifiers and ICIs to promote the permanent functional activation of T cells is a fairly new combination method to achieve a sustained antitumor immune response [117].

Moreover, complex environmental factors—such as changes in cell metabolism, the accumulation of metabolites, nutrient supply, and various enzyme actions—have a great impact on the mechanism of ICI resistance within the TME [118]. In exploring antitumor immunotherapy means, we can only focus on a certain main line and continuously optimize it to produce the best antitumor efficacy to reduce the drug-resistant population of patients to the greatest extent. For ICIs, how to permanently and effectively activate the immune system, enhance the immune microenvironment, and produce better antitumor efficacy in the form of positive feedback is an urgent problem to be solved.

## 4. Therapeutic Strategies for Overcoming Drug Resistance with ICIs

As research progress and clinical ICIs have shown some limitations in recent years, there is a growing sense that the efficacy of single drug use cannot be guaranteed [119,120]. Combination therapy has become an important safeguard against ICI resistance. In view of the possible factors of ICI drug resistance mentioned above, this section summarizes and proposes several strategies and guidelines for overcoming ICI drug resistance (Figure 2).

### 4.1. The Enhanced Destruction of T Cells

There are three essential conditions required for T cells to produce antitumor efficacy: high infiltration, the strong activation of T cells, and correct recognition of cancer cells. It is only when the three conditions are met at the same time that T cells can have strong functions for tumor destruction. However, the inhibitor alone does not produce all three effects on effector immune cells, but merely blocks tumor inhibition of T cells for ICIs. Hence, to boost the clinical effectiveness of ICIs, it is necessary to develop new synergistic treatment methods to reduce the drug resistance of ICIs more effectively.

#### 4.1.1. Recruiting T Cells

Cytokines play a crucial role in the antitumor response [121]. Factors that recruit T cells and boost immune cell activity are sometimes even more important for clearing ICI-resistant patients. First, chemokines are key factors in promoting T-cell recruitment [122,123]. Dangaj et al. (2019) found that the chemokines CCL5 and CXCL9 are associated with T-cell infiltration in solid tumors. CCL5 is produced by tumor cells, while CXCL9 is produced by TAMs and DCs, which are dependent on antigen recognition and IFN-γ-specific induction. The authors found that the co-expression of CCL5 and CXCL9 could promote CD8^+^ T cells infiltration, prolong survival, and alleviate reactivity to PD-1 inhibitors in tumor tissue [124]. In addition, IFN-γ can enhance the activity of CD8^+^ T cells and the immune response of TH1 cells, and inhibit the function of other immunosuppressive cells such as Tregs, MDSCs, and TAMs. Meanwhile, IFN-γ can have an antitumor cell proliferation effect and promote cell apoptosis [125]. Gao et al. (2016) found that ipilimumab-resistant melanoma patients had defects in IFN-γ pathway-related genes or had obvious amplification of IFN-γ pathway-inhibiting genes [126]. Similarly, Zaretsky et al. found defects in the IFN-γ pathway gene JAK1/2 in melanoma patients who relapsed after an initial response to pembrolizumab and developed acquired resistance [127]. The importance of IFN-γ signaling pathway defects in anti-CTLA-4 and anti-PD-1 checkpoint therapy was emphasized. Further, type I interferons (IFN-α and IFN-β) also play a substantial role in immune regulation. For example, activators of the STING pathway can stimulate type I IFN to play a role in antitumor immunity, which has been confirmed in a number of studies [128,129,130]. Moreover, Tang et al. (2016) showed that LIGHT (tumor necrosis factor superfamily member 14, TNFSF14) also plays a role in T-cell recruitment. By binding to lymphotoxin B receptor (LTβR), the expression of chemokines and adhesion molecules is upregulated, and CD8^+^ T cells are recruited. Tumor regression in mice treated with TNFSF14, in combination with PD-L1 inhibitors, was complete and resulted in long-term immune memory [131].

The rise of bispecific antibodies (biAbs), especially T-cell bispecific antibodies, provides the potential to further enhance the antitumor activity of immune cells. T-biAbs can bind TCRs and tumor-specific antigens simultaneously, and bypass the restrictions of the MHC peptide complex. The T-cell binding arm of T-biAbs usually uses the CD3ε antibody, which can effectively recruit specific T cells and significantly mitigate the cytotoxic effect [132]. Blinatumomab is the only FDA-approved T-biAb and is a CD3/CD19 biAb for the treatment of relapsed or refractory B-cell progenitor acute lymphoblastic leukemia (B-ALL) patients [133]. In addition to blinatumomab, other bispecific antibodies targeting T cells have been undergoing clinical trials in recent years. Pillarisetti et al. (2020) proposed a GPRC5D/CD3 biAb (JNJ-64407564), and data showed that it could recruit CD3^+^ T cells to target and kill GPRC5D-positive multiple myeloma (MM) tumor cells [134]. However, in solid tumors, the destruction by T-biAbs of solid tumors needs to be verified due to its concentration limitation. For example, Li et al. (2018) found that T-biAbs of HER2/CD3 recruited increased T-cell infiltration and enhanced recognition in a murine breast cancer model with HER2 overexpression [135]. In addition, Bartlett et al. conducted a phase Ib trial (NCT02665650) of CD30/CD16A biAb (AFM13) in combination with the PD-1 inhibitor pembrolizumab for relapsed or refractory Hodgkin’s lymphoma. CD30 is a specific target of Hodgkin cells, and the CD16A receptor is a key factor in NK cells and macrophages. The results of this study confirmed that the combination therapy of CD30/CD16A biAb and the PD-1 inhibitor has superior clinical and pharmacodynamic activity, as well as a positive recruitment effect on immune cells [136].

#### 4.1.2. Improving the Specific Recognition of T Cells

Chimeric antigen receptor (CAR) is synthesized in vitro and can make T cells recognize tumors more efficiently. Currently, CAR-T-cell therapy is the most important research achievement using CAR receptors, attaining immune clearance through the infusion of large amounts of autologous or allogeneic antitumor immune cells into tumor patients. In contrast to T-cell recruitment factors, CAR-T immunotherapy eliminates the limitation of low immune cell infiltration in some tumors by injecting large numbers of highly activated cytotoxic T-cells to clear tumor tissue. For example, the infusion of CAR-T-cells into patients with chronic myeloid leukemia who have relapsed after hematopoietic stem cell transplantation (HSCT), or patients with metastatic melanoma, has been shown to achieve partial or complete remission [137,138,139]. Moreover, researchers indicated that engineered T cells combining CD19 and CAR receptors can escape the restriction of binding TCR and MHC peptides, and broaden the ability of T cells to bind to cell surface antigens. In a clinical trial of CD19-chimeric CAR-T-cell therapy in adults with relapsed and refractory diffuse large B-cell lymphoma (NCT02445248) in 2021, the authors reported an overall response rate of 53% in 115 patients who received treatment at a median follow-up of 40.3 months. Thirty-nine percent of patients had complete remission, which showed lasting activity and controllable safety compared with traditional therapy [140]. More importantly, the combination of CAR-T-cells and ICIs could precisely meet the three conditions for T cells at the same time, enabling ICIs to achieve more comprehensive tumor cell destruction. In recent studies, CAR-NK cells, CAR-M cells, and other methods have emerged [141,142,143,144]. All of these strategies not only attempt to overcome the unique immunotoxicity of CAR-T-cells, but also provide more options for CAR receptor therapy strategies, which will complement ICI therapy. In turn, this may produce greater significance for helping overcome the tumor resistance of ICIs.

### 4.2. Improving Immunogenicity in Tumor Tissue

Tumor-associated antigens (TAAs), important T-cell activation, and tumor-recognition molecules play an important role in the reversal of the entire immune microenvironment [145]. However, for most tumors, there is often a very low concentration of tumor-associated antigens in the tumor environment, which results in insufficient T cells being activated by ICI drugs to produce tumor-specific killing effects through the antigens, ultimately reducing the therapeutic effect of ICIs. For tumor cells, the recognition sites of intermediate antigens are usually located on the inner surface of the cell membrane. During ICI drug therapy, T cells are unable to recognize tumor cell antigens effectively, which causes another mechanism of immune resistance. Improving the immunogenicity of the tumor microenvironment to increase T cells’ targeting and recognition of tumor cells is another vital strategy for synergistic ICI drugs.

The most common idea for releasing intracellular antigens is to destroy some initial tumor cells to release antigens, and to activate DCs and T cells to form a positive feedback effect. The high immunogenicity of the tumor microenvironment makes ICI drugs have a better play space, which means a better synergistic antitumor effect. Since this idea emerged, how to efficiently induce tumor cells to produce immunogenic death (ICD) and release more immune antigens has become a compensation mechanism for ICI clinical drug resistance. Tumor cells that undergo ICD release not only TAAs but also DAMPs, such as high-mobility group box 1 (HMGB1), extracellular ATP, calreticulin (CALR), type I IFN, and heat shock protein (HSP), which are recognized by pattern recognition receptors (PRRs) expressed in myeloid and lymphoid cells, afterward inducing immune responses and establishing immune memory [102,103,146]. In this section, we will introduce different strategies for inducing ICD in tumor cells.

#### 4.2.1. The Biodestructive Effect

The release of TAAs, through the killing of tumor cells by oncolytic viruses (OVs), is also a crucial research focus. OVs reproduce themselves and mediate antitumor activity through a unique biological mechanism. OVs can preferentially replicate in tumor cells rather than normal cells, directly causing lysis of tumor cells to release TAAs and promote the ICD process for tumor cells. OVs can also repeat the lysis process by infecting neighboring tumor cells, and can inhibit tumor angiogenesis, regulate the tumor microenvironment (TME), and recruit T cells [147]. Talimogene laherparepvec (T-VEC) is the first OV approved and active in malignant melanoma. Based on herpes simplex virus type 1 (HSV-1), T-VEC eliminates the ICP34.5 (infected cell protein 34.5) encoding gene to enhance tumor selectivity, and eliminates the ICP47 encoding gene to increase tumor antigen presentation. Furthermore, the virus inserts the GM-CSF gene to promote the recruitment and maturation of APCs [148]. Many clinical trials have been conducted to investigate the efficacy of OVs in combination with ICIs in the treatment of tumors. For example, Chesney et al. (2018) reported the results of a phase II clinical trial in combination with T-VEC and the anti-CTLA-4 antibody (ipilimumab) in patients with advanced, unresectable melanoma. The objective response rate of the combination group was 39%. Compared to the 18% objective response rate of the single drug ipilimumab [149], the rate was significantly higher. In addition, clinical trials based on OVs may offer promising treatment prospects for gliomas with poor response to conventional ICI treatment [150].

#### 4.2.2. The Physical/Chemistry-Destructive Effect

ICIs combined with physical therapy are the most common method to improve efficacy. Radiotherapy is a traditional adjuvant therapy that is very common in clinical applications. Radiotherapy cannot only kill tumors, but also significantly increase immunogenicity in the tumor microenvironment after tumor destruction. This undoubtedly creates the possibility for the combination therapy of radiotherapy and ICIs [151]. For example, Theelen et al. (2021) tried to treat non-small-cell lung cancer (NSCLC) with pembrolizumab with or without radiotherapy. The results showed that the best abscopal response rate (ARR), the best abscopal disease control rate (ACR), the median progression-free survival rate, and the median overall survival rate in patients treated with pembrolizumab alone were lower than those in the pembrolizumab plus radiotherapy group. This suggests that radiotherapy significantly improved the response and prognosis of pembrolizumab in patients with NSCLC [152].

Photodynamic therapy (PDT) is another physical method to induce ICD and promote the immune response in tumor cells. Photosensitizers such as porphyrins or non-porphyrins can selectively accumulate in tumor tissues and generate cytotoxic substances in the presence of oxygen through visible light irradiation at specific wavelengths, leading to ICD in tumor cells. At the same time, DAMP release from induced dead tumor cells initiates a tumor adaptive immune response, and different photosensitizers induce ICD in different ways [153]. Duan et al. (2016) proposed that Znpyrophosphate (ZnP) nanoparticles, loaded with the photosensitizer Pyrolipid (ZnP@pyro), involved strategies with PD-L1 inhibitors to treat PD-L1-resistant 4T1 tumor models in mice. They found that combination therapy was effective in eradicating the primary tumor and preventing pulmonary metastasis, as well as inhibiting existing distant metastases [154]. Lan et al. (2018) also confirmed that PDT enhanced the efficacy of PD-L1 therapy and had a distant effect in a mouse model of colorectal cancer, with tumor regression >90% [155].

Moreover, hyperthermia is an important means of inducing ICD, among which radiofrequency (RF) ablation is the primary hyperthermia method that has entered clinical practice. Shi et al. (2016) found that the combined application of RF ablation and PD-1 inhibitors could significantly enhance the antitumor effect of T cells and prolong survival [156]. Domingo-Musibay et al. (2017) demonstrated increased plasma HSP70 levels in patients with advanced melanoma following RF ablation. They also tried to use RF therapy combined with granulocyte-macrophage colony-stimulating factor (GM-CSF) for tumor therapy. However, this clinical trial did not show the therapeutic advantage of RF ablation in combination with GM-CSF [157]. The strategy of RF ablation combined with immunotherapy still needs further exploration. With the progress of nanotechnology, light-sensitive and heat-sensitive nanomaterials have become emerging ICD induction methods, which have greater advantages in good tumor targeting and modifiability [158], thus greatly improving the efficiency of ICD induction. The combination of physical therapy and ICIs may usher in new breakthroughs.

Chemotherapeutics (such as anthracyclines, DNA-damage agents, and mitotic poisons) and some targeted antitumor agents (such as tyrosine kinase inhibitors) can also induce ICD [159]. Pfirschke et al. (2016) conducted a study on lung adenocarcinoma models. They found that oxaliplatin-cyclophosphamide (OXA-CYC) treatment could significantly enhance HMGB1 staining in KP tumors (KRAS and Trp53 mutations) and directly activate the upregulation of toll-like receptor 4 (TLR4) expression in dendritic cells and macrophages, further increasing the ratio of CD8^+^ T cells to Tregs. A significantly improved antitumor efficacy effect was observed compared to the mouse model using the anti-PD-1 antibody and the anti-CTLA-4 antibody alone [160]. On the other hand, chemotherapeutics can boost the antitumor immune effect in other ways. For example, Alizadeh et al. (2014) found an impact in addition to the ICD effect in tumor cells after Adriamycin chemotherapy. In their trial, Adriamycin selectively promoted the apoptosis of MDSCs in the spleen, blood, and tumor bed; inhibited the immunosuppressive activity of residual MDSCs; and increased the proportion of CD4^+^ T cells, CD8^+^ T cells, NK cells, and their downstream effector factors [161]. However, the toxicities and side effects of chemotherapy, and the uncertainty in the application of combined ICIs, are still the limitations of this scheme; a more systematic, comprehensive evaluation of the combined antitumor efficacy of chemotherapy and ICIs needs to be carried out in depth.

#### 4.2.3. A Tumor Vaccine

Therapeutic cancer vaccines are prepared from tumor-associated antigens (TAAs) or tumor-specific antigens (TSAs) by delivering these antigens directly to specialized antigen-presenting cells such as DCs or macrophages. Vaccines can directly stimulate the immune system and induce an immune response, thus activating CD4^+^ and CD8^+^ T cells to kill tumors [36,79]. There are many kinds of cancer vaccines, including cell, nucleic acid, protein polypeptide, and genetically engineered vaccines [162]. Regardless of the type of vaccine, the underlying principle is to present tumor antigens directly to immune cells and to activate the immune response, thus achieving an anticancer effect. To some extent, adoptive cellular immunotherapy (ATT) is a major cellular vaccine strategy; it provides a new method of tumor-specific antigen delivery. When immune cells have a low antigen load or cannot obtain the relevant tumor antigen load [81], this scheme can achieve better tumor targeting and killing effects by directly expressing tumor cell antigen antibodies in immune cells [83,163]. Ott et al. (2017) conducted a clinical trial and found that six melanoma patients who received personalized vaccines showed significant tumor regression. During the median follow-up period of 25 months, four patients did not experience recurrence, and two patients relapsed after subsequent treatment with PD-1 inhibitors [164]. Hu et al. (2021) published the results of a clinical trial on the long-term memory T-cell response in melanoma patients after receiving a personalized tumor vaccine.

During the 4-year follow-up period of eight patients with melanoma after receiving a personalized long peptide vaccine, all of the patients survived, and six of them had no evidence of disease activity. In addition, vaccine-induced CD4^+^ T cells were detected in vitro to have a memory phenotype, which can produce a continuous T-cell response [165]. In research on tumor vaccines, the selection of antigens is crucial. However, specific antigens still cannot be effectively selected, so how to prepare TSAs with excellent targeting tumor cells has become the focus of current research.

### 4.3. Other Therapeutic Strategies for Overcoming ICI Resistance

#### 4.3.1. The Combination of ICIs Themselves

Although the use of ICI tumor therapy has been promoted by the development process, the use of a single ICI cannot achieve an ideal response rate or be influenced by other immunosuppressive factors in the TME or other mechanistic pathways related to ICIs. Moreover, multiple immune targets are often co-expressed on the same cell surface, which drives the current research on combination therapy. Combination therapy includes two or more ICIs used in combination, or immune inhibitors combined with local radiotherapy, chemotherapy, tyrosine kinase inhibitors, tumor vaccines, immune activation agents, and stimulant agonists [166]. Combination therapy aims to maximize the antitumor T-cell effect, and to neutralize metabolic and immunosuppressive factors in the TME via different mechanisms of action to reduce drug resistance and achieve the persistence of antitumor effects. Among them, the combined use of ICIs has become a research hotspot. In 2015, a phase III trial (CheckMate 067) was proposed for nivolumab and ipilimumab in combination for advanced melanoma. The median progression-free survival (mPFS) and objective response rate (OR) were 11.5 months and 57.6% in the combination group, 6.9 months and 43.7% in the single-drug nivolumab group, and 2.9 months and 19.0% in the single-drug ipilimumab group [167], which showed a better antitumor effect in the ICI combination group. Furthermore, Wolchok et al. (2017) proposed that ICI combination therapy could improve the 3-year overall survival (OS) and progressive-free survival (rPFS) of patients with BRAF mutation, M1c staging, and elevated lactate dehydrogenase levels. In both the single-drug nivolumab group and the single-drug ipilimumab group, the OS and rPFS were much lower [14]. Hence, the combination strategy of ipilimumab and nivolumab provides promising new treatment options for lung cancer (including relapsed SCLC and advanced NSCLC) and advanced renal cell carcinoma [168,169,170]. Because the combination therapy of ICIs has a predictable therapeutic effect, other combinations of ICIs are being explored.

#### 4.3.2. Epigenetic Regulation

Epigenetic regulation has become an emerging hotspot in tumor immunotherapy [171,172]. It was found that in the same population of T cells, different epigenetic states resulted in different levels of T-cell activation and sensitivity regarding related signaling pathways. How to regulate surface genetic states (such as methylation levels) to make the immune system attain a state of functional activation sensitivity is another emerging collaborative approach to improve the efficacy of ICIs. Peng et al. (2015) demonstrated that enhancing the zeste homolog 2 (EZH2) and DNA methyltransferase 1 (DNMT1) inhibitors can induce tumor contraction and increase TIL by constructing a human ovarian cancer mouse tumor model. This can promote the efficacy of PD-L1 inhibitors [173]. In 2021, researchers found that the loss of methyltransferase SETDB1 and methylase demethylase KDM5B could also trigger immune stimulation. The loss of KDM5B induced strong antitumor immunity in a mouse melanoma model. T cells, especially CD8^+^ T cells, are more invasive in terms of boosting the efficacy of ICIs. KDM5B also recruits SETDB1 to exert immunosuppressive effects [174,175]. Therefore, epigenetic modifiers that disrupt the stability of KDM5B or SETDB1, or interfere with the interaction of KDM5D and SETDB1, may have a synergistic antitumor effect with ICIs.

## 5. Conclusions and Future Development of Immune Checkpoint Inhibitors

With the rapid development of immune checkpoint inhibitors and their entry into clinical settings, good therapeutic effects have been widely used in the treatment of various tumors. However, a large amount of data indicates that significant resistance still exists in some patients treated with ICIs against cancer. The efficacy of ICIs may vary according to ICI regimens and tumor types. To date, ICI response rates have rarely exceeded 40% [176]. Due to the heterogeneity of tumors, the ICI drug resistance mechanism is complicated, interlaced, and dynamic, and diverse schemes to overcome the drug resistance mechanism have different degrees of efficacy in individual patients. Hence, the focus of ICI antitumor immunotherapy is to further clarify the specific drug resistance pathway of ICIs to explore universal tumor biomarkers that are responsive to ICI application, and to obtain the best immune response through the rational application of ICIs.

ICIs directly target the immunosuppressive effect between tumor tissues and the immune system by specifically blocking immune checkpoints, preventing immune escape of tumor tissues, and activating the cytotoxic function of endogenous immune cells. ICI immunotherapy strategies have been widely used to treat metastatic melanoma, renal cell carcinoma, and lung cancer. Compared to traditional chemoradiotherapy, ICIs have developed rapidly in clinical practice, with visible survival advantages and low toxicity side effects. However, due to the complex mechanism of ICI action, the cross-reactivity of signaling pathways, and the metabolic effects of the immune microenvironment, the clinical response to ICIs in some tumor patients is weak. In addition to ICI resistance, immunotherapy-induced toxicity (e.g., T-cell-mediated cytokine release syndrome, T-cell-mediated neurotoxicity, etc.) is another major reason for the limited clinical use of ICIs. The overproduction of immune storms by activated immune cells can be fatal for cancer patients, and while ICIs hold great promise, their significant toxic side effects are still somewhat poorly studied.

To solve ICI drug resistance and specific immunotoxicity, immune checkpoint inhibitors may be preferred in combination therapy. Compared to ICIs alone, combination therapy with ICIs is more conducive to overcoming ICI resistance caused by various complex factors in view of the characteristics of the high heterogeneity of the tumor microenvironment to compensate for deficiencies. As immunotherapy matures, more targeted immunotherapy strategies will emerge, and with the appearance of more immune targets, the clinical drug selection for ICIs will become richer. In the future, for cancer patients, the challenge of immunotherapy may be more about how to combine ICIs with other immune-antitumor strategies to comprehensively and effectively activate the immune response, and to generate more powerful potential antitumor ability.

## Figures and Tables

**Figure 1 cancers-14-03575-f001:**
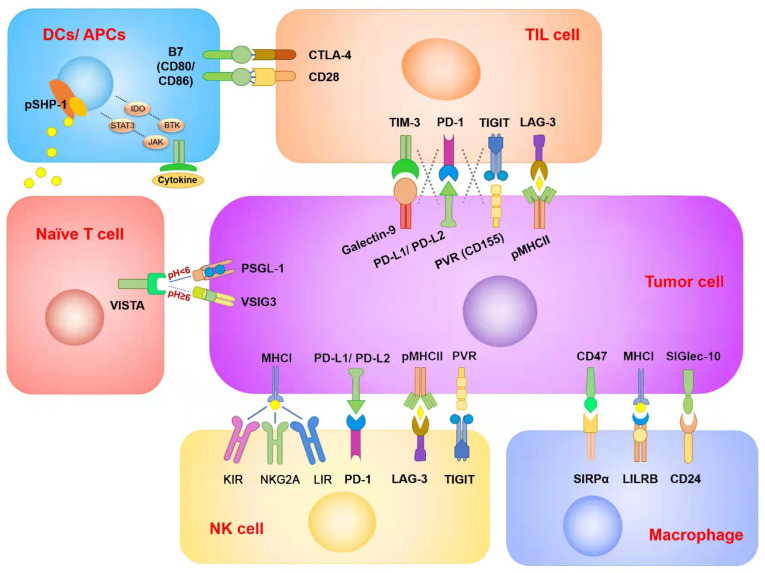
Summary of immune checkpoints in different immune cells and tumor cells. Each ICI was targeted to different cell types. Although two of the ICIs’ target receptors and corresponding ligands are in the same checkpoint signaling pathway, there are also differences in therapeutic efficacy and side effects between them. In addition, there may be overlapping inhibitory effects between different checkpoint signaling pathways in the same immune cell type. This is one of the reasons for ICI resistance.

**Figure 2 cancers-14-03575-f002:**
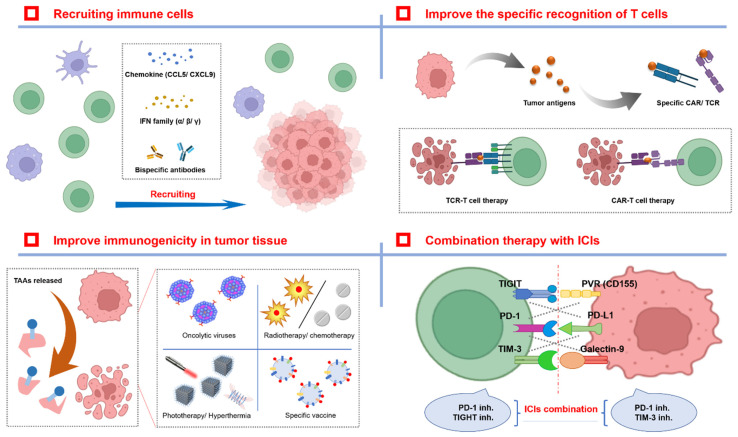
Different therapeutic strategies for overcoming drug resistance with ICIs. There are two main strategies for improving ICI therapy effectiveness and reducing the number of patients with ICI resistance. The first strategy is to improve immune cell infiltration in TME by some cytokines or chemokines, and enhance T cell recognition for tumor cells. The second strategy is to increase the concentration of tumor antigens in THE TME by biological, physical, or chemical methods to facilitate antigen presentation by APC. In addition, the combination of different ICI types may also produce synergistic antitumor effects.

**Table 1 cancers-14-03575-t001:** Clinical trials of different kinds of ICIs for immunotherapy.

Strategy	Targeted Cell	Drug Name	Trail Type	Targeted Tumor Type	Trail Identifier
PD-1 inh.	CD8^+^ T cell	Nivolumab *	Phase 2 study	Hodgkin lymphoma, Lymphoma	NCT02181738
PD-1 inh.	CD8^+^ T cell	Pembrolizumab *	Phase 2 study	HNSCC, Lip SCC, Oral cavity cancer	NCT03082534
CTLA-4 inh.	CD8^+^ T cell	Ipilimumab *	Phase 2 study	Melanoma, NT	NCT02743819
CTLA-4 inh.	CD8^+^ T cell	Quavonlimab (MK-1308)	Phase 1/2 study	Bronchial neoplasms, NSCLC	NCT03179436
TIM-3 inh.	CD8^+^ T cell	Sabatolimab (MBG453)	Phase 1/2 study	NSCLC, RC, Melanoma	NCT02608268
LAG-3 inh.	CD8^+^ T cell	Ieramilimab (LAG525)	Phase 1/2 study	NSCLC, RC, Mesothelioma	NCT02460224
TIGIT inh.	CD8^+^ T cell	Vibostolimab (MK-7684)	Phase 1 study	Neoplasms, Gastric cancer, NSCLC	NCT02964013
VISTA inh.	Naive T cell	CI-8993	Phase 1 study	ST	NCT04475523
SIRPα inh.	Macrophage	CC-95251	Phase 1 study	Hematologic neoplasms	NCT03783403
SIRPα inh.	Macrophage	BI 765063	Phase 1 study	ST	NCT03990233
LILRB inh.	Macrophage	JTX-8064	Phase 1/2 study	Advanced refractory ST malignancies	NCT04669899
Akt inh.	DC	Capivasertib (AZD5363)	Phase 2 study	Metastatic breast cancer	NCT02423603
IDO inh.	DC	Indoximod	Phase 1/2 study	Neoplasms, NT	NCT02073123
KIR inh.	NK cell	IPH2101	Phase 1 study	AML patients over the age of 60	NCT01256073
NKG2A inh.	NK cell	Monalizumab	Phase 2 study	HNSCC	NCT02643550
PD-L1 inh.	Tumor cell	TQB2450	Phase 2 study	ESCC	NCT05038813
PD-L1 inh.	Tumor cell	Atezolizumab *	Phase 3 study	Non-squamous NSCLC, Squamous NSCLC	NCT2409342
PD-L2 vac.	Tumor cell	PD-L2 peptide	Phase 1 study	Follicular lymphoma	NCT03381768
CD47 inh.	Tumor cell	Evorpacept (ALX148)	Phase 1 study	ST malignancy, Relapsed or refractory non-Hodgkin lymphoma	NCT03013218
PVR inh.	Tumor cell	NTX-1088	Phase 1 study	Advanced malignant ST	NCT05378425
B7(CD80/CD86)	Tumor cell	Z-CTls	Phase 1 study	NSCLC	NCT03060343

Abbreviations—inh.: inhibition.; vac.: vaccination; DC: dendritic cell; AML: acute myeloid leukemia; HNSCC: Head and neck squamous cell carcinoma; NT: neuroectodermal tumors; NSCLC: non-small cell lung cancer; RC: renal carcinoma; ST: solid tumor; ESCC: esophageal squamous cell carcinoma. *: ICIs drugs that FDA approved. Indications: Ipilimumab: unresectable or metastatic melanoma; adjuvant treatment of melanoma; advanced renal cell carcinoma; microsatellite instability-high or mismatch repair deficient metastatic colorectal cancer; hepatocellular carcinoma; metastatic non-small cell lung cancer; malignant pleural mesothelioma; esophageal cancer. Nivolumab: unresectable or metastatic melanoma; adjuvant treatment of melanoma; neoadjuvant treatment of resectable non-small cell lung cancer; metastatic non-small cell lung cancer; malignant pleural mesothelioma; advanced renal cell carcinoma; classical Hodgkin lymphoma; head and neck squamous cell carcinoma; urothelial carcinoma; microsatellite instability-high or mismatch repair deficient metastatic colorectal cancer; hepatocellular carcinoma; esophageal cancer; gastric cancer, gastroesophageal junction cancer, and esophageal adenocarcinoma. Pembrolizumab: melanoma; non-small cell lung cancer; head and neck squamous cell carcinoma; classical Hodgkin lymphoma; primary mediastinal large B-cell lymphoma; urothelial carcinoma; microsatellite instability-high or mismatch repair deficient cancer; microsatellite instability-high or mismatch repair deficient colorectal cancer; gastric cancer; esophageal cancer; cervical cancer; hepatocellular carcinoma; Merkel cell carcinoma; renal cell carcinoma; endometrial carcinoma; tumor mutational burden-high cancer; cutaneous squamous cell carcinoma; triple-negative breast cancer. Atezolizumab: urothelial carcinoma; non-small cell lung cancer; small cell lung cancer; hepatocellular carcinoma; melanoma.

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
