# Peer review of "Immune Checkpoint Inhibitors in Cancer Therapy—How to Overcome Drug Resistance?"

_cancers, 2022, doi:10.3390/cancers14153575_

Round 1

Reviewer 1 Report

This review summarizes and classifies different types of immune checkpoints inhibitors (ICI) for anticancer therapy, their cellular and molecular targets and action mechanisms. Then authors overview the main reasons of resistance to ICI and suggest possible solutions to overcome it. The review is well organized, include recent information and nice figures. This topic would be of interest to researchers and clinical oncologists.

But I have some observations:

From my point of view, it would be more logical to classify the mechanisms of the antitumor action of ICIs basing not on immune cell types, but on signaling axes (for example, ICIs for the PD-1/PD-L1/2 axis). In this case, therapeutic strategies are easier to explain. In addition, some ICIs target different cell types: for example, not only CD8+ cells but also NK cells express PD-1 (please check Table 1 and Figure 1 carefully). The authors did not pay enough attention to NK cells as targets for ICI therapy, this should be corrected.

Author Response

Reviewer 1

This review summarizes and classifies different types of immune checkpoints inhibitors (ICI) for anticancer therapy, their cellular and molecular targets and action mechanisms. Then authors overview the main reasons of resistance to ICI and suggest possible solutions to overcome it. The review is well organized, include recent information and nice figures. This topic would be of interest to researchers and clinical oncologists.

 Response:

We greatly appreciate your comments, and we hope our article could help more tumor patients with ICIs resistance in clinical.

But I have some observations:

From my point of view, it would be more logical to classify the mechanisms of the antitumor action of ICIs basing not on immune cell types, but on signaling axes (for example, ICIs for the PD-1/PD-L1/2 axis). In this case, therapeutic strategies are easier to explain. In addition, some ICIs target different cell types: for example, not only CD8+ cells but also NK cells express PD-1 (please check Table 1 and Figure 1 carefully). The authors did not pay enough attention to NK cells as targets for ICI therapy, this should be corrected.

Response:

Thank you for pointing out this critical issue.

Firstly, we greatly agree with the point that “it would be more logical to classify the mechanisms of the antitumor action of ICIs basing not on immune cell types, but on signaling axes”.

The signaling axis such as PD-1/PD-L1, is a signaling pathway to achieve tumor immunosuppression. Both of PD-1 and PD-L1 play an essential role in this inhibition axis. Using different signal axes to classify immunosuppression is also the classification method of most current papers. However, after a large number of literature investigation, we gradually found that two different molecular inhibitors of the same signaling axis such as PD-1/PD-L1 would have different antitumor effects and toxic side effects. Therefore, we hypothesized that the two molecular inhibitors may have different mechanisms of action due to different immune cells, even if they inhibit the same signaling axis. And that's the reason that we classify the mechanisms by ICIs in different immune cells.

In addition, NK cell is actually also another key ICIs targets for immune cells. After reflecting on the review, we added some important researches on ICIs for NK cell immunosuppression in the part of 2.1.4.

Reviewer 2 Report

This article highlights the various antitumor action sites and drug resistance mechanisms associated with immune checkpoint inhibitors (ICI). I find this review informative and well-written. The language of communication is elegant. The illustrations made by the authors are appealing and exciting. In particular, the figure that explained different therapeutic strategies to overcome ICI resistance. Brilliant. Congratulations to the authors on drafting this work well.

I have a few minor comments to highlight.

Although, I found table 1 useful, I would encourage the authors to briefly mention briefly about ICIs that are already FDA approved and their respective clinical indications.

Can the authors explain the limitations associated with ICI in detail? For instance, what is the maximum clinical target response rate (%) achieved by different cancer patients and what are the autoimmune disorders (names and their %) due to ICI usage? That will be very informative.

When its comes to resistance mechanisms with patients with mutations at IFN/JAK or mismatch repair proficient, does the author recommend any particular therapeutic strategy to overcome ICI resistance? Please discussion this briefly.

I enjoyed reading this manuscript. I highly recommend this work for publication.

Author Response

Reviewer 2

This article highlights the various antitumor action sites and drug resistance mechanisms associated with immune checkpoint inhibitors (ICI). I find this review informative and well-written. The language of communication is elegant. The illustrations made by the authors are appealing and exciting. In particular, the figure that explained different therapeutic strategies to overcome ICI resistance. Brilliant. Congratulations to the authors on drafting this work well. 

Response:

We greatly appreciate your comments and high recognition of the article. We hope our article could help more tumor patients with ICIs resistance in clinical.

 I have a few minor comments to highlight.Although, I found table 1 useful, I would encourage the authors to briefly mention briefly about ICIs that are already FDA approved and their respective clinical indications. Can the authors explain the limitations associated with ICI in detail? For instance, what is the maximum clinical target response rate (%) achieved by different cancer patients and what are the autoimmune disorders (names and their %) due to ICI usage? That will be very informative. When it comes to resistance mechanisms with patients with mutations at IFN/JAK or mismatch repair proficient, does the author recommend any particular therapeutic strategy to overcome ICI resistance? Please discussion this briefly.I enjoyed reading this manuscript. I highly recommend this work for publication. 

Response: We greatly thank for the opinions and thanks again for your recognition of our work.

Firstly, as you suggested, clinical indications are important for different ICIs that are already FDA approved for cancerous person, because of the specificity for different targets. We have summarized all indications of ICIs drugs that FDA-approved in Table 1.

Secondly, it is important to discuss the limitations of ICIs with reported values. After literature research, we found that the clinical response rates of immunotherapy are not too high. The remission rate of patients treated with CTLA-4 inhibitor is about 15%, and the remission rate of PD-1/PD-L1 inhibitor rarely exceeds 40%. In addition, the occurrence of immune-related adverse events (irAEs) is another major reason for the limited clinical application of ICIs such as dermatologic toxicity (maculopapular rash, pruritis, etc.), gastrointestinal toxicity(diarrhea, colitis and hepatitis, etc.), endocrinopathy (thyroid toxicity, pituitary toxicity, etc.), pneumonitis, rheumatologic toxicity (arthritis, polyarthritis, etc.), neurologic toxicity (myasthenia gravis, noninfectious encephalitis/myelitis, etc.), renal toxicity(acute kidney injury, etc.), ocular toxicity(uveitis, peripheral ulcerative keratitis, etc.), cardiovascular toxicity(myocarditis, pericarditis, etc.) and hematologic toxicity(hemolytic anemia, red cell aplasia, etc.). The information has also been added in the introduction of the article.

Finally, as for IFN/JAK, deletion of IFN-γ/JAK pathway genes are often associated with ICIs treatment resistance. Some people found that the silencing of JAK1 or JAK2 in a variety of tumor targets could enhance the sensitivity of tumor cells to NK cell-mediated lysis. And IFN-γ could induced activation of JAK1, JAK2 and STATE, leading to up-regulation of PD-L1 expression and further inducing NK cell resistance. That is to say, ICIs with IFN-γ/JAK pathway would inhibit tumor growth by NK cells. For IFN-γ/JAK signaling pathway ICIs-resistant patients, the activation level of NK cells may be the main reason for resistance because NK cells could resist tumors without antigen recognition. Therefore, ICIs targeting NK cells such as monalizumab to further enhance NK cell activity, or in combination with PD-L1 inhibitors to promote the anti-tumor effect of T cells may have synergistic effects.

Mismatch repair proficient (pMMR) tends to indicate a low tumor mutation burden (TMB) in tumor microenvironment. So there usually be little tumor-associated antigen (TAA) to activate immunity. Therefore, in this state, how to use external methods such as chemotherapy, phototherapy, and oncolytic virus to improve the immunogenic death of tumor cells may be the key method to solve ICIs resistance.

Round 2

Reviewer 1 Report

The authors have responded appropriately to criticisms. They made appropriate changes to the manuscript. I believe that the revised manuscript can be published and will make a worthy contribution to this area of biomedical research.